# A Novel bHLH Transcription Factor PtrbHLH66 from Trifoliate Orange Positively Regulates Plant Drought Tolerance by Mediating Root Growth and ROS Scavenging

**DOI:** 10.3390/ijms232315053

**Published:** 2022-11-30

**Authors:** Beibei Liang, Shiguo Wan, Qingling Ma, Li Yang, Wei Hu, Liuqing Kuang, Jingheng Xie, Yingjie Huang, Dechun Liu, Yong Liu

**Affiliations:** Department of Pomology, College of Agronomy, Jiangxi Agricultural University, Nanchang 330045, China

**Keywords:** citrus, drought stress, bHLH transcription factor, gene function, ROS scavenging, root growth

## Abstract

Drought limits citrus yield and fruit quality worldwide. The basic helix-loop-helix (bHLH) transcription factors (TFs) are involved in plant response to drought stress. However, few bHLH TFs related to drought response have been functionally characterized in citrus. In this study, a bHLH family gene, named *PtrbHLH66*, was cloned from trifoliate orange. PtrbHLH66 contained a highly conserved bHLH domain and was clustered closely with bHLH66 homologs from other plant species. PtrbHLH66 was localized to the nucleus and had transcriptional activation activity. The expression of *PtrbHLH66* was significantly induced by polyethylene glycol 6000 (PEG6000) and abscisic acid (ABA) treatments. Ectopic expression of *PtrbHLH66* promoted the seed germination and root growth, increased the proline and ABA contents and the activities of antioxidant enzymes, but reduced the accumulation of malondialdehyde (MDA) and reactive oxygen species (ROS) under drought stress, resulting in enhanced drought tolerance of transgenic Arabidopsis. In contrast, silencing the *PtrbHLH66* homolog in lemon plants showed the opposite effects. Furthermore, under drought stress, the transcript levels of 15 genes involved in ABA biosynthesis, proline biosynthesis, ROS scavenging and drought response were obviously upregulated in *PtrbHLH66* ectopic-expressing Arabidopsis but downregulated in *PtrbHLH66* homolog silencing lemon. Thus, our results suggested that PtrbHLH66 acted as a positive regulator of plant drought resistance by regulating root growth and ROS scavenging.

## 1. Introduction

Drought stress is one of the major environmental factors limiting plant growth, development and yield in semi-arid and arid areas [1]. During a long-term evolutionary process, plants have developed a complex mechanism at the physiological, biochemical and molecular levels in response to drought stress [2,3]. As core components in the drought signal transduction pathway, transcription factors (TFs) play an important role in regulating plant response to drought stress by activating or limiting the transcription of downstream target genes [4].

The basic helix-loop-helix (bHLH) proteins constitute the second-largest TF family in plants [5]. The bHLH TFs contain a highly conserved bHLH domain (approximately 50–60 amino acids in length), consisting of a basic region at its N-terminus and an HLH region at the C-terminus. The basic region is composed of 10–15 residues, typically rich in basic amino acids, and functions as a DNA binding domain to recognize and specifically bind the E-box (5′-CANNTG-3′) or G-box (5′-CACGTG-3′) in the promoter of target genes. The HLH domain contains approximately 40–50 amino acids, including two amphipathic α-helices linked by a loop with variable amino acid length. This domain can promote protein–protein interactions to form homodimers or heterodimers with other bHLH proteins [6]. To date, a large number of bHLH family genes has been identified from various plant species, including Arabidopsis [7], poplar [8], rice [9], apple [10], grapes [11], peach [12], tomato [13], pear [14] and so on. The bHLH TFs have been reported to act as important regulators involved in the regulation of various plant developmental and metabolic processes, such as nodule vascular patterning [15], root hair formation [16], stomata and root development [17,18], trichome initiation [19], leaf senescence [20], seed germination [21], flowering regulation [22], hormone metabolism [23,24], biosynthesis of alkaloid, nicotine, tryptophan [25], flavonoid and anthocyanin [26,27]. In addition, previous reports have revealed that bHLH TFs also play important roles in regulating plant responses to drought and other abiotic stresses. For example, Arabidopsis AtbHLH122 acts as a positive regulator of plant response to drought, NaCl and osmotic stresses by repressing *CYP707A3* expression and increasing abscisic acid (ABA) content [28]. Overexpression of tomato *SlbHLH22* increases the contents of secondary metabolites and osmolytes, enhances reactive oxygen species (ROS) scavenging capacity and improves drought and salt tolerance in tomato [13]. Wheat TabHLH49 plays a positive role in regulating wheat resistance to drought stress by mediating the transcript of the dehydrin *WZY2* gene [29]. Ectopic expression of apple *MdbHLH130* enhances tobacco drought resistance by regulating stomatal closure and ROS scavenging [30]. Apple MdCIB1 positively regulates plant tolerance to drought stress [31].

Citrus is one of the most economically important fruit crops in the world. However, its growth, development, fruit yield and quality are severely limited by drought stress [32]. It is estimated that up to 82% of the potential yield in citrus is lost due to drought and other abiotic stresses [33]. To date, several bHLH TFs have been identified in citrus. For instance, *Clemenules mandarin* CitbHLH1 makes a great contribution in regulating abscission-related events in citrus abscission zones [34]. *Satsuma mandarin* CubHLH1 is involved in the control of carotenoid metabolism [35]. PtrICE1 and PtrbHLH from trifoliate orange (*Poncirus trifoliata* (L.) Raf.) play an important role in plant response to cold stress [36,37,38]. However, studies on bHLH TFs involved in regulating drought tolerance are limited in citrus. Trifoliate orange is one of the most commonly used citrus rootstocks in China. Therefore, it is important to identify the drought-responsive genes from trifoliate orange to improve its drought tolerance by transgenic technology. Several genes, such as *PtrABF*, *PtrMAPK*, *PtADC*, *PtsrMYB*, *PtrNAC72*, *PtrZPT-1* and *PtrCDPK10*, have been reported to be involved in trifoliate orange response to drought stress [39,40,41,42,43,44,45]. However, most of the drought-responsive genes are still unclear in trifoliate orange. Recently, we screened out many differentially expressed genes (DEGs) from trifoliate orange under normal growth conditions and drought stress using transcriptome sequencing. In these DEGs, the expression of a bHLH family gene, named *PtrbHLH66*, was significantly upregulated under drought stress, suggesting it might play an important role in trifoliate orange tolerance to drought stress. In the present study, we cloned and characterized *PtrbHLH66* from trifoliate orange. A further study revealed that *PtrbHLH66* plays a positive role in mediating plant tolerance to drought stress through regulating root growth and ROS scavenging. Our results provided new insight into the molecular mechanism of citrus response to drought stress.

## 2. Results

### 2.1. Sequence Analysis of PtrbHLH66

In this study, a novel bHLH family gene, named PtrbHLH66, was cloned from trifoliate orange and deposited in the NCBI GenBank database under accession number OP009586. The open reading frame (ORF) of PtrbHLH66 was 1380 bp in length and encodes a bHLH family protein of 459 amino acids, with a putative molecular weight of 48.32 kDa and an isoelectric point of 5.93. BLAST analysis showed that PtrbHLH66 shared high identities with bHLH66 proteins from other plant species, including Citrus sinensis CsbHLH66 (97.62%, GenBank accession number XP_006473971.1), Malus domestica MdbHLH66 (56.96%, XP_028964276.1), Arabidopsis thaliana AtbHLH66 (52.90%, BAD44153.1) and Oryza sativa OsbHLH66 (48.64%, XP_015627343.1). Multiple sequence alignment analysis revealed that PtrbHLH66 contained a conserved bHLH domain (Figure 1A). To study the relationship between PtrbHLH66 and its homologous proteins from other plants, a phylogenetic tree was constructed based on the neighbor-joining algorithm with Poisson model of MEGA X software. The result revealed that PtrbHLH66 was first clustered with citrus sinensis bHLH66 and then classified with Pistacia vera bHLH66-like, Mangifera indica RHL1-like, Mangifera indica RHL1 and carica papaya bHLH66-like. The Zea mays RHL1 and Oryza sativa bHLH66 from monocots were more genetically distant from PtrbHLH66 than other bHLH66 homologues (Figure 1B). A total of 10 motifs was detected in PtrbHLH66 and its homologous proteins using MEME analysis, of which motif 1 (HLH region), motif 3 (basic region), motif 6 and motif 8 were shared by all bHLH66 proteins (Figure 1C).

### 2.2. PtrbHLH66 Is a Nucleus Protein That Has Transcriptional Activation Activity

To verify the subcellular localization of PtrbHLH66, the coding sequence (CDS) of *PtrbHLH66* was fused with green fluorescent protein (GFP) under the control of a cauliflower mosaic virus 35S (CaMV 35S) promoter to produce recombinant plasmid (35S:PtrbHLH66-GFP). The recombinant plasmid and positive control (35S:GFP) were separated into the leaf epidermal cell of tobacco. Microscopic visualization showed that GFP fluorescence from the fusion protein was only detected in the nucleus, whereas the fluorescence in the positive control was observed in the cytoplasm and nucleus, indicating that PtrbHLH66 is localized in the nucleus (Figure 2A). Furthermore, the transcriptional activation activity of PtrbHLH66 was investigated by a yeast two-hybrid experiment. The CDS of PtrbHLH66 was fused with a GAL4 DNA-binding domain in pGBKT7 to generate recombinant plasmid pGBKT7-PtrbHLH66. The treatment (pGBKT7-PtrbHLH66+ pGADT7), positive control (PGBKT7-53+pGADT7-T) and negative control (PGBKT7+ pGADT7) were, respectively, introduced into the yeast cell Y2H Gold. The yeast cells transformed with pGBKT7-PtrbHLH66 or positive control grew normally on SD/-Trp-His-Ade-Leu medium and showed blue color on SD/-Trp-His-Ade-Leu/X-α-gal medium. In contrast, yeast cells harboring negative control did not grow on both SD/-Trp-His-Ade-Leu and SD/-Trp-His-Ade-Leu/X-α-gal media (Figure 2B). These results suggested that PtrbHLH66 exhibited transcriptional activation activity in yeast cells.

### 2.3. Expression Analysis of PtrbHLH66 in Trifoliate Orange after Polyethylene Glycol 6000 (PEG6000) and ABA Treatments

The expression levels of *PtrbHLH66* in the roots and leaves of trifoliate orange were determined by real-time quantitative PCR (qRT-PCR) after PEG6000 and ABA treatments. The expression levels of *PtrbHLH66* in the roots were strongly induced after 1 h of PEG6000 treatment, peaked at 3 h and then decreased to a level still much higher than that at the start of the treatment. In the leaves, the *PtrbHLH66* transcript levels were increased and peaked at 1 h of PEG6000 treatment and then decreased gradually from 1 h to 24 h (Figure 3A). After 1 h of ABA treatment, the transcript levels of *PtrbHLH66* in the roots and leaves were increased to the maximum level and then declined afterwards (Figure 3B). Notably, the expression levels of *PtrbHLH66* in the roots were significantly higher than those in the leaves after PEG6000 and ABA treatments (Figure 3A,B).

### 2.4. Ectopic Expression of PtrbHLH66 Enhances Drought Stress Resistance in Arabidopsis by Promoting Root Growth and ROS Accumulation

In this study, transgenic Arabidopsis ectopic-expressing *PtrbHLH66* was produced to further investigate the function of *PtrbHLH66*. A total of 15 positive T2 transgenic Arabidopsis lines after kanamycin selection was further confirmed by leaf genomic PCR (Appendix A). The transcript levels of *PtrbHLH66* in two positive lines (35S-3 and 35S-8) were significantly higher than those in the wild-type (WT) plants, which were selected for further study (Figure 4A).

To investigate the effect of drought stress and ABA on seed germination, the seeds of the WT and two transgenic Arabidopsis lines were sown on the MS medium (control) and MS medium supplemented with 6% PEG6000 or 0.5 μM ABA. There was no significant difference in seed germination rates between the two genotypes under control conditions. However, the seed germination rates of the two transgenic lines were much higher than those of the WT plants after 6% PEG6000 or 0.5 μM ABA treatment (Figure 4B,C). Furthermore, the 7-day-old seedlings of two transgenic lines possessed significantly longer primary root and lateral root and much more lateral roots in comparison to the WT seedlings, both under control conditions and after 6% PEG6000 or 0.5 μM ABA treatment (Figure 4D–G).

The 3-week-old seedlings of the two genotypes cultivated in soil were used to further study the function of *PtrbHLH66* in response to drought stress. After 3 weeks of drought treatment, much more severe leaf wilting was observed in the WT plants compared with the two transgenic lines (Figure 5A). Moreover, the two transgenic lines showed significantly higher survival rates in comparison to the WT plants under drought stress (Figure 5B). Meanwhile, the contents of malondialdehyde (MDA), proline and ABA were similar between the WT and two transgenic lines under normal growth conditions. After drought treatment, the two transgenic lines exhibited significantly lower contents of MDA and much higher levels of proline and ABA compared with the WT plants (Figure 5C–E).

To study the effect of drought stress on ROS production, the accumulation of superoxide anion radical (O^2−^) and hydrogen peroxide (H_2_O_2_) in leaves from the WT and transgenic lines was investigated by nitroblue tetrazolium (NBT) and 3,3′-diaminobenzidine (DAB) staining, respectively. The leaves of two transgenic lines showed much weaker NBT and DAB staining compared to WT leaves after drought stress (Figure 5F), suggesting the two transgenic lines accumulated less ROS after drought treatment. This result was further confirmed by the fact that less H_2_O_2_ contents were observed in the two transgenic lines under drought stress (Figure 5G). In addition, the activities of superoxide dismutase (SOD), peroxidase (POD) and catalase (CAT) in the leaves of the two transgenic lines were much higher than those in the WT plants after drought treatment (Figure 5H–J), indicating that ectopic expression of PtrbHLH66 reduces ROS accumulation by increasing the antioxidant enzyme activities. These results suggest that the transgenic plants are much more tolerant to drought stress in comparison to the WT plants.

### 2.5. Silencing of PtrbHLH66 Homolog Decreases Drought Tolerance in Lemon (Citrus limon) Plants by Reducing Root Growth and ROS Accumulation

To investigate the role of *PtrbHLH66* in citrus response to drought stress, tobacco rattle virus (TRV)-based virus-induced gene silencing (VIGS) was used to silence the *PtrbHLH66* homolog in lemon. The positive *PtrbHLH66* homolog silencing lemon plants were confirmed by genomic PCR (Appendix A). The control lemon plants were transformed with the TRV2 empty vector and named TRV2. The expression levels of the *PtrbHLH66* homolog in two VIGS lemon lines, defined as VIGS-2 and VIGS-7, were significantly reduced in comparison to the TRV2 plants (Figure 6A). Under normal growth conditions, no morphological differences were observed between the TRV2 and VIGS lemon plants. However, the TRV2 plants exhibited more severe leaf wilting in comparison to the two VIGS lemon lines after drought treatment (Figure 6B). Meanwhile, no significant differences in the total root lengths, total root surface areas, total root volumes, primary root lengths, lateral root numbers and lateral root lengths were detected between the TRV2 and VIGS lemon plants under normal growth conditions. However, these values in the two VIGS lemon lines were much lower than those in the TRV2 plants after drought treatment (Figure 6C–H).

In addition, the TRV2 and VIGS lemon plants displayed negligible differences in the contents of leaf MDA, proline and ABA under normal growth conditions. However, the contents of proline and ABA in the leaves of two VIGS lines were much lower than those of TRV2 plants under drought stress. In contrast, the leaf MDA content in the two VIGS lines was significantly higher than that in the TRV2 plants (Figure 7A–C). The leaf discs of TRV2 and VIGS plants were similarly and lightly stained by NBT and DAB under normal growth conditions. The staining in both genotypes became darker after drought treatment. Notably, the VIGS leaf discs were stained deeper than the TRV2 leaf discs under drought stress (Figure 7D). Meanwhile, the leaves of two VIGS lines exhibited a significantly higher content of H_2_O_2_ and much lower activities of SOD, POD and CAT in comparison to the TRV2 leaves, indicating that the leaves of VIGS plants accumulated much more ROS than did TRV2 plant leaves by reducing the antioxidant enzyme activities (Figure 7E–H). These results suggested that the two VIGS lemon lines were much more susceptible to drought stress than the TRV2 plants.

### 2.6. PtrbHLH66 Positively Regulates the Expression of Drought-Related Genes under Drought Stress

To further study the molecular mechanism of PtrbHLH66 in the regulation of plant drought tolerance, the expression levels of 15 drought-related genes involved in ABA biosynthesis (*AAO3*, *ABA2*, *NCED3*, *NCED9*, *ZEP*), ROS scavenging (*SOD*, *POX* and *CAT*), proline biosynthesis (*P5S5*) and drought stress response (*DREB1A*, *DREB2A*, *DREB3*, *RD20*, *RD29*, *EDR1*) were examined in the leaves of the WT, transgenic Arabidopsis ectopic-expressing *PtrbHLH66*, TRV2 and VIGS lemon plants by qRT-PCR. The transcript levels of most genes in the two transgenic Arabidopsis lines were similar to the WT plants under the normal growth condition, with the exception of *AtRD29A*, whose expression in the two transgenic lines was significantly higher than that in the WT plants. After drought treatment, the expression levels of all 15 genes were much higher in the two transgenic Arabidopsis lines than those in the WT plants (Figure 8A). In contrast, the two VIGS lemon lines possessed much lower expression of all 15 genes in comparison to the TRV2 plants under drought stress (Figure 8B). These results suggested that PtrbHLH66 played a positive role in regulating the expression of drought-related genes under drought stress.

## 3. Discussion

Citrus is an economically important fruit crop throughout the world. The yield and quality of citrus fruits highly depends on sufficient water supply. However, as a perennial woody plant, citrus often suffers drought stress, especially in dried and semi-dried areas, leading to great economic loss. For improving citrus drought tolerance, it is critical to study the molecular mechanism of citrus response to drought stress.

To date, many bHLH TFs have been reported to be involved in plant response to drought stress [9,13,31]. However, the contribution of bHLH TFs to drought tolerance in citrus is still unknown. In the present study, a novel bHLH gene, called *PtrbHLH66*, was cloned from trifoliate orange. Bioinformatic analysis revealed that PtrbHLH66 had a conserved bHLH domain in its C terminus and possessed high identities with other bHLH66 homologous proteins (Figure 1A). The evolutionary analysis showed that PtrbHLH66 was closely clustered with its homologous proteins, such as *Citrus sinensis* bHLH66, *Pistacia vera* bHLH66-like, *Mangifera indica* RHL1-like, *Mangifera indica* RHL1 and *Carica papaya* bHLH66-like (Figure 1B). These results indicated that PtrbHLH66 belonged to the citrus bHLH family.

Previous reports revealed that bHLH TFs were localized in the nucleus and could activate the expression of downstream genes by binding the E-box or G-box in the promoter sequences of target genes [6,28]. Similarly, our results showed that PtrbHLH66 was a nuclear localized protein and had transcriptional activation ability in yeast cells (Figure 2A,B). However, further studies must be carried out to determine which promoter sequences can be bound by PtrbHLH66 to activate the expression of downstream genes in citrus.

It has been reported that the expression levels of many plant bHLH genes, including cotton *GhbHLH1* [46], poplar *PebHLH35* [47], wheat *TabHLH1* [48], tartary buckwheat *FtbHLH3* [49] and maize *ZmPTF1* [18], increased after PEG and ABA treatments. In agreement with these reports, the expression of *PtrbHLH66* in leaves and roots of trifoliate orange was also induced by PEG and ABA treatments (Figure 3A,B), suggesting that the *PtrbHLH66* might play an important role in citrus response to drought stress via an ABA-dependent pathway.

Increasing studies show that bHLH TFs are involved in regulating plant tolerance to drought stress [28,29,31]. In this paper, ectopic expression of *PtrbHLH66* improved the drought tolerance in Arabidopsis, as indicated by the less plant damage and significantly higher survive rates under drought stress (Figure 5A,B). In contrast, much more severe wilting leaves were observed in VIGS lemon plants after drought treatment (Figure 6B), suggesting the silence of the *PtrbHLH66* homolog reduced the lemon tolerance to drought stress. These results indicated that *PtrbHLH66* positively regulated plant tolerance to drought stress.

To understand the mechanism of *PtrbHLH66* regulating plant response to drought stress, a series of morphological, physiological and biochemical changes were investigated in the *PtrbHLH66* ectopic-expressing Arabidopsis and VIGS lemon. Under drought stress, the seed germination rate usually changed in response to the osmotic stress caused by water deficiency [50]. In this study, the transgenic Arabidopsis ectopic-expressing *PtrbHLH66* exhibited much higher seed germination rates compared with the WT plants after PEG6000 treatment (Figure 4B,C), indicating that *PtrbHLH66* could increase seed germination rate to cope with drought stress.

Root is the major organ that absorbs water and nutrients in plants, thereby playing an important role in protecting the plant against drought stress [51]. Several bHLH TFs were reported to improve drought tolerance by regulating root development. For example, Arabidopsis AtbHLH68 modulates lateral root elongation and plant tolerance to drought stress [52]. Ectopic expression of *MdCIB1* increases root length and improves drought tolerance in Arabidopsis [31]. Consistent with these reports, our results revealed that ectopic expression of *PtrbHLH66* promoted the primary and lateral root growth in Arabidopsis under drought stress (Figure 4D–G). In contrast, silence of the *PtrbHLH66* homolog limited the primary and lateral root growth in lemon plants after drought treatment (Figure 6B–H). These results suggested that *PtrbHLH66* positively regulated root growth to improve plant drought tolerance.

Under drought stress, excessive ROS mainly include H_2_O_2_ and O^2−^, accumulated in the plant, leading to severe damage to organelles by oxidizing lipids, proteins and DNA in plant cells [53]. MDA, a product of membrane lipid peroxidation, is usually used to measure the degree of plant membrane–lipid peroxidation [54]. In this study, histochemistry staining and quantitative measurement assays revealed that the ROS and MDA levels were obviously reduced in the leaves of *PtrbHLH66* ectopic-expressing Arabidopsis (Figure 5C,F,G), while they were significantly increased in the VIGS lemon leaves under drought stress (Figure 7A,D,E), suggesting *PtrbHLH66* negatively regulated ROS accumulation and oxidative injury in plant cells. A set of antioxidant enzymes, such as SOD, POD and CAT, plays an important role in ROS scavenging and protecting plant cells against oxidative damage [55]. Our results showed that the activities of three antioxidant enzymes (SOD, POD and CAT) were remarkably increased in the *PtrbHLH66* ectopic-expressing Arabidopsis leaves (Figure 5H–J), but obviously decreased in the leaves of VIGS lemon after drought treatment (Figure 7F–H). Accordingly, the expression levels of three antioxidant enzyme genes (*SOD*, *POX* and *CAT*) were significantly upregulated in the transgenic Arabidopsis, but obviously downregulated in the VIGS lemon under drought stress (Figure 8A,B). These results suggested that *PtrbHLH66* positively regulated the transcript of antioxidant enzyme genes to enhance antioxidant enzyme activities, leading to improvements in ROS scavenging capacity and a decrease in oxidative injury under drought stress.

Proline functions as an important osmolyte to regulate the plant response to drought stress [56]. The *P5CS* gene encodes a key rate-limiting enzyme involved in proline biosynthesis [57]. Tartary buckwheat *FtbHLH3* regulates plant drought tolerance by upregulating the *P5CS* transcript to increase proline accumulation [49]. Consistent with this report, ectopic expression of *PtrbHLH66* increased the *AtP5CS* expression and proline contents in the leaves of Arabidopsis after drought treatment (Figure 5D and Figure 8A). In contrast, significant declines were detected in the *ClP5CS* expression and proline levels of the VIGS lemon leaves under drought stress (Figure 7B and Figure 8B). Therefore, it could be speculated that *PtrbHLH66* positively mediated *P5CS* gene expression and proline accumulation in plant to keep the osmotic balance between the intracellular and extracellular environments, thereby influencing cellular membrane integrity and plant tolerance to drought stress.

ABA is one of the most important phytohormones known to regulate plant response to drought stress [58]. Several bHLH genes, such as Arabidopsis At*bHLH122* [28], *AtbHLH112* [59], *AtbHLH68* [52], wheat *TabHLH1* [48], tartary buckwheat *FtbHLH3* [49], tomato *SlbHLH22* [13] and *Myrothamnus flabellifolia MfbHLH38* [60], have been reported to regulate plant drought tolerance through an ABA-dependent pathway. Our results showed that ectopic expression of *PtrbHLH66* increased the ABA contents in the leaves of Arabidopsis under drought stress (Figure 5E). However, silence of the *PtrbHLH66* homolog in lemon led to a significant decline in leaf ABA levels after drought treatment (Figure 7C). Moreover, the seed germination rates, primary root and lateral root lengths and lateral root numbers in the *PtrbHLH66* ectopic-expressing Arabidopsis were much higher than those in the WT plants after ABA treatments (Figure 4B–G). The 9-cisepoxycarotenoid dioxygenase (NCED) catalyzes the cleavage of 9-cis-epoxycarotenoids, which is the rate-limiting step in ABA biosynthesis [61]. A two-step conversion of zeaxanthin to alltrans-violaxanthin during ABA biosynthesis was catalyzed by a zeaxanthin epoxidase (ZEP) [62]. ABA deficient 2 (ABA2) encodes a xanthoxin dehydrogenase responsible for catalyzing the conversion of xanthoxin to abscisic aldehyde during ABA biosynthesis [63]. The final step in ABA biosynthesis was catalyzed by aldehyde oxidase 3 (AAO3) [64]. qRT-PCR results revealed that the expression levels of *NCED3*, *NCED9*, *ZEP*, *ABA2* and *AAO3* were significantly upregulated in the *PtrbHLH66* ectopic-expressing Arabidopsis but downregulated in the VIGS lemon under drought stress (Figure 8A,B), indicating that *PtrbHLH66* could positively regulate the expression of genes involved in ABA biosynthesis. In addition, the expression of *PtrbHLH66* was induced by ABA (Figure 3B). These results suggested that *PtrbHLH66* might play an important role in regulating plant drought tolerance via an ABA-dependent pathway.

*DREB1A*, *DREB2A*, *DREB3* and *ERD1* genes were involved in regulating plant drought tolerance through ABA-independent pathways [65,66,67,68]. *RD20* is an ABA-responsive gene involved in plant response to drought stress [69]. *RD29A* is a drought-responsive gene mediated by both ABA-dependent and ABA-independent pathways [70,71]. The expression levels of *DREB1A*, *DREB2A*, *DREB3*, *ERD1*, *RD29A* and *RD20* were upregulated in the *PtrbHLH66* ectopic-expressing Arabidopsis but downregulated in the VIGS lemon under drought stress (Figure 8A,B), suggesting the *PtrbHLH66* might positively regulate drought-related gene expression by both ABA-dependent and ABA independent pathways. However, further studies should be performed to study which genes could be directly regulated by PtrbHLH66 and whether PtrbHLH66 is involved in ABA-dependent or/and ABA independent pathways.

## 4. Materials and Methods

### 4.1. Plant Materials

The seeds of trifoliate orange obtained from an orchard of Xinfeng County, Ganzhou City, Jiangxi province in China, were soaked in 2% sodium hypochlorite solution for 20 min, then rinsed three times with anhydrous ethanol for 30 s at each time and finally washed three times with double-distilled water. The sterilized seeds were sown in MS medium and placed in a culture room (25 °C, 16 h light/8 h dark in a day, 80% relative humidity). Germinated seeds were transplanted to pots filled with soil (garden soils, peats and sands mixed at the ratio of 3:2:1), irrigated with 200 ml water every three days, fertilized by compound fertilizer (N ≥ 15%, K_2_O ≥ 15%, P_2_O_5_ ≥ 15%) every 15 days and grown to a height of 28 ± 2 cm after three-month cultivation. Then, the seedlings were used for gene cloning and expression analysis.

### 4.2. Cloning and Bioinformatics Analysis of PtrbHLH66

The cDNA sequence of *bHLH66* homologues from trifoliate orange (Pt1g019480) was obtained from the trifoliate orange genomic database (http://citrus.hzau.edu.cn/, accessed on 15 October 2022). The CDS of *PtrbHLH66* was amplified by PCR with a pair of gene-specific primers (Forward: 5′ ATGCAAGGAATCAGCTCGCTC 3′; Reverse: 5′ TCAGGGCTTGGAAACGGAAGC 3′). Total RNA was isolated from the third and fourth young leaves (5 to 6 days old) of trifoliate orange by a MiniBEST RNA extraction kit (TaKaRa, Dalian, China). The integrity and quality of total RNA were checked by 1% agarose gel electrophoresis and a Nano Drop 2000 spectrophotometer (Thermo Fisher Scientific, Waltham, USA). The M5 Sprint qPCR RT kit with gDNA remover (Mei5 Biotechnology Co., Ltd., Beijing, China) was used for cDNA synthesis. The process of *PtrbHLH66* clone, including PCR, PCR product purification, ligation, transformation and sequencing, was carried out according to our previous report [72]. The bioinformatics analyses, including protein sequence analysis, blast analysis, multiple sequence alignment and phylogenetic analysis were performed according to the methods of our previous report [73]. The meme tool (https://meme-suite.org/meme/tools/meme, accessed on 15 October 2022) was used to analyze the protein motif of PtrbHLH66.

### 4.3. Expression Analysis of PtrbHLH66 in Trifoliate Orange

Before stress treatments, trifoliate orange seedlings were transferred to Hoagland’s solution in an artificial climate chamber at 25 °C with 80% relative humidity (16 h light/8 h dark in a day) for 5 days to adapt the seedlings to novel conditions. For drought and ABA treatments, the seedlings were transferred to Hoagland’s solution containing 20% PEG6000 and 100 µM ABA, respectively. After 0, 1, 3, 6, 12 and 24 h of PEG6000 and ABA treatments, the leaves and roots from trifoliate orange seedlings were collected for expression analysis. For each sample per biological replicate, leaves and roots were harvested randomly from 10 seedlings. Three biological replicates were carried out for each time point of each treatment.

Transcript levels of *PtrbHLH66* in leaves and roots of trifoliate orange under drought and ABA stresses were carried out by qRT-PCR. A pair of *PtrbHLH66*-specific primers (Forward: 5′ GCATCGGGTTGCCTTTAT 3′; Reverse: 5′ AGCCTGCGTCTGGTTCAT 3′) was used for expression analysis. Trifoliate orange *ACTIN* gene (Forward: 5′ TGGTTCAACTATGTTCCCTG 3′; Reverse: 5′ ACTCATCATACTCGCCCTTT 3′) was used as endogenous control to normalize the transcript levels among different samples. The qRT-PCR was performed according to our previous report [73].

### 4.4. Subcellular Localization and Transcriptional Activation Analysis

The CDS of *PtrbHLH66* was cloned into pBWA(V)HS:GFP vector (35S:GFP) using a pair of primers (Forward: 5′ GAGAACACGGGGGACTCTAGAATGCAAGGAATCAGC TCGCT 3′; Reverse: 5′ CCATGGTACCCCCGGGGATCCGGGCTTGGAAACGGAAG 3′) to construct recombinant plasmid 35S:PtrbHLH66-GFP. The 35S:GFP (positive control) and 35S:PtrbHLH66-GFP plasmids were transformed into *Agrobacterium tumefaciens* GV31011 and then introduced into the leaves of 3-week-old tobacco. After 48 h cultivation in darkness at 28 °C, the transformed leaf cells were observed under SP5 confocal laser scanning microscope (Leica, Wetzlar, Germany).

To examine the transcriptional activation activity of PtrbHLH66, the CDS of *PtrbHLH66* was cloned to the pGBTKT7 vector using a pair of primers (Forward: 5′ CGCG AATTCATGCAAGGAATCAGCTCGCT 3′, underline represents *Eco*R I site; Reverse: 5′ GCGGGATCCTCAGGGCTTGGAA ACGGAAG 3′, underline represents *Bam*H I site) to generate the recombinant plasmid of pGBKT7-PtrbHLH66. The pGBKT7-PtrbHLH66 and pGADT7 were co-transformed into Y2HGold yeast cells. Two pairs of plasmids (pGBKT7 and pGADT7, pGBKT7-53 and pGADT7-T) were also co-transformed into Y2HGold yeast cells to produce negative control and positive control, respectively. Then, the transformed colonies were selected on SD/-Trp-His, SD/-Trp-His-Ade-Leu and SD/-Trp-His-Ade-Leu/X-a-gal media.

### 4.5. Generation of Arabidopsis Ectopic-Expressing PtrbHLH66

The CDS of *PtrbHLH66* was cloned from trifoliate orange leaves by PCR using a pair of *PtrbHLH66*-specific primers with *Xba* I and *Sac* I restriction sites (Forward: 5′ GCGT CTAGAATGCAAGGAATCAGCTCGCTC 3′, underline represents *Xba* I site; Reverse: 5′ CGCGAGCTCTCAGGGCTTGGAAACGGAAGC 3′, underline represents *Sac* I site). The restriction endonuclease *Xba* I and *Sac* I was used to digest both the PCR product and pBI121 vector. Then, the two enzyme digestion products were ligated by a T4 DNA ligase (TaKaRa, Dalian, China) to construct the recombinant plasmid of pBI121-PtrbHLH66 driven by the CaMV35S promoter. To generate Arabidopsis ectopic-expressing *PtrbHLH66*, the pBI121-PtrbHLH66 was introduced into the Arabidopsis using *Agrobacterium tumefaciens*-mediated (strain GV310) floral dipping method as described by a previous report [74]. Positive transgenic Arabidopsis plants were screened on MS medium containing 50 mg⋅L^−1^ kanamycin and 30 mg⋅L^−1^ hygromycin and further confirmed by genomic PCR with a pair of *PtrbHLH66*-specific primers (Forward primer: 5′ ATGCAAGGAATC AGCTCGCTC 3′; Reverse primer: 5′ TCAGGGCTTGGAA ACGGAAGC 3′). The transcript of *PtrbHLH66* in leaves of the WT and T2 transgenic Arabidopsis plants was detected by qRT-PCR with the PtrbHLH66-specific primers (Forward: 5′ GCATCGGGTTGCCTTTAT 3′; Reverse: 5′ AGCCTGCGTCTGGTTCAT 3′). Arabidopsis *AtACTIN* gene (Forward: 5′ GAAACCCTCGTAGATTGGCA 3′; Reverse: 5′ CTCTCCCGCTATGTATGTCGC 3′) was used as the endogenous control to normalize the transcript levels among different samples. The positive T2 transgenic Arabidopsis lines were used for further experiments.

### 4.6. Generation of VIGS Lemon Plants

The VIGS assay was carried out according to a previous report [75]. In brief, the 300 bp CDS fragment of *PtrbHLH66* was cloned by a pair of primers (Forward: 5′ CGCGGATCCTCATTCAAGTCCCAACAGGG 3’, underline represents *Bam*H I site; Reverse: 5′ GCG GAGCTCAATTCTCT CTCTGCGCAATC 3′, underline represents *Sac* I site). This fragment was introduced into the *Bam*H I and *Sac* I restriction sites of tobacco TRV2 to produce the recombinant plasmid of pTRV2-PtrbHLH66. The pTRV2-PtrbHLH66, pTRV1 and pTRV2 (negative control) vectors were introduced into Agrobacterium tumefaciens GV3101 using the freeze–thaw method. The GV3101 solution of pTRV2-PtrbHLH66 and pTRV2 was mixed with pTRV1, respectively, at a volume ratio of 1:1. The mixed solutions were used to infiltrate lemon-germinating seeds with 2 cm emerging shoots in a vacuum chamber at −20 kPa (twice, each time for 50 s). Afterwards, the seeds were rinsed with water, dried on filter papers and transferred to plastic pots with soil in a normal growth condition (25 °C,16 h light/8 h dark in a day, 80% relative humidity). Genomic PCR with the primers (Forward: 5′ ATTCACTGGGAGATGATACGCT 3′; Reverse: 5′ GAGCTCAATTCTCTCTCTGCGC 3′) was performed to screen the positive VIGS lemon plants. The expression levels of PtrbHLH66 homologous gene in VIGS lemon plants were detected by qRT-PCR using the primers (Forward: 5′ CAGCAA GGTAGGGT TA 3′; Reverse: 5′ GAGGGTCTCAGGTAGTTC 3′).

### 4.7. Analysis of Seed Germination Rates, Survival Rates and Root

For measurement of seed germination rates under drought and ABA stresses, the sterilized seeds of the WT and transgenic Arabidopsis plants ectopic-expressing *PtrbHLH66* were sown on MS medium (control) and MS medium supplemented with 6% PEG6000 and 0.5 μM ABA for 5 days. To investigate the root growth condition under drought and ABA stresses, the seeds of the two genotypes were grown on the same MS media for 7 days. The root images were obtained by Epson Expression 10000XL scanner (Epson, Long Beach, USA) and analyzed by WinRHIZOTM root analysis system (Regent, Quebec, Canada) to obtain data for the primary root lengths, later root lengths and later root numbers. For drought treatment, 3-week-old plants of the WT and transgenic Arabidopsis grown in soil were deprived of water for 3 weeks and used to calculate the survival rates. Meanwhile, 5-week-old plants of the TVR2 (infiltrated with pTRV1 and pTRV2) and VIGS lemon plants were subjected to water withholding for 3 weeks. The root images of Arabidopsis and lemon plants were obtained by Epson Expression 10000XL scanner (Epson, Long Beach, USA) and analyzed by WinRHIZOTM root analysis system (Regent, Quebec, Canada). Three biological replicates were carried out for the above experiments.

### 4.8. Histochemical Staining and Physiological Index Measurements

Histochemical staining of NBT and DAB was used to monitor the O^2−^- and H_2_O_2_ accumulation, respectively, in Arabidopsis and lemon leaves [76]. The contents of H_2_O_2_, MDA, proline and the activities of SOD, POD and CAT in Arabidopsis and lemon leaves were detected by corresponding measurement kits (Nanjing Jiancheng Bioengineering Institute, Nanjing, China) according to a previous report [77]. The ABA contents in Arabidopsis and lemon leaves were measured by enzyme-linked immunosorbent assay (ELISA) according to a previous report [78]. Three biological replicates were used for the above experiments.

### 4.9. Expression Analysis of the Drought-Stress-Related Genes in Transgenic Plants

After drought treatment described as above, the expression levels of 15 drought-related genes were measured in the leaves of Arabidopsis and lemon plants by qRT-PCR with gene specific primers (Appendix A). The leaves of Arabidopsis and lemon plants were sampled for total RNA extraction as described above. Arabidopsis *AtACTIN* and lemon *ClACTIN* genes were used as endogenous control to normalize the expression levels among Arabidopsis and lemon samples, respectively (Appendix A). The qRT-PCR was performed as the method described in our previous report [73]. Three biological replicates were used for qRT-PCR analysis.

### 4.10. Statistical Analysis

All data in the present study are shown as the means ± standard deviations (SD) of three biological replicates. Statistical analyses are performed by Student’s t-test in SPSS version 22. Statistical significance was shown as * (*p* < 0.05) and ** (*p* < 0.01).

## 5. Conclusions

In summary, this paper reported the isolation and characterization of a drought- and ABA-responsive gene, *PtrbHLH66,* from trifoliate orange. Ectopic expression and gene silencing assays revealed that *PtrbHLH66* positively regulated the expression levels of genes involved in ABA biosynthesis, proline biosynthesis, ROS scavenging and drought response, resulting in changes in ABA, proline, ROS and MDA contents, root growth and antioxidant enzyme activities under drought stress and, finally, positively mediating plant resistance to drought stress. The results of this study not only improve our understanding of the regulatory mechanism of citrus response to drought stress, but also provide a novel candidate gene for drought-tolerance breeding in citrus.

## Figures and Tables

**Figure 1 ijms-23-15053-f001:**
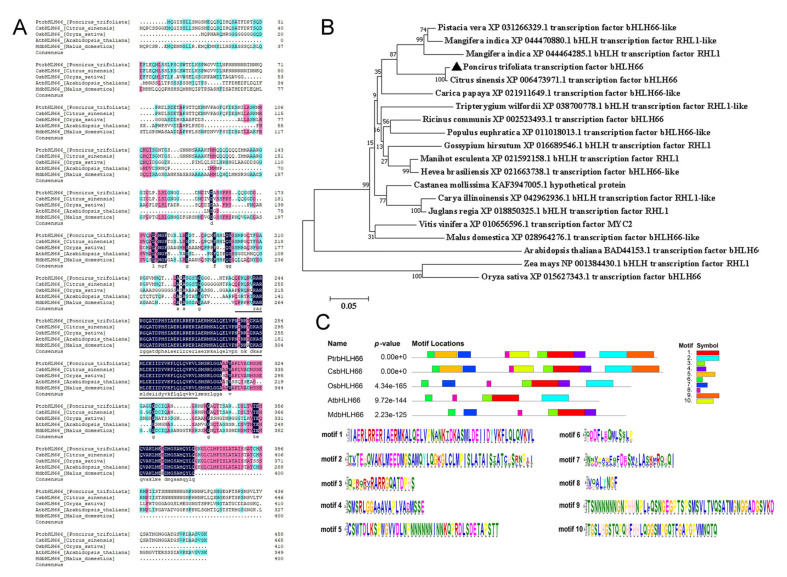
Structural and phylogenetic analysis of the PtrbHLH66 protein. (**A**) Multiple sequence alignment of PtrbHLH66 with its homologous proteins from different plant species, including *Citrus sinensis* CsbHLH66 (GenBank accession number: XP_006473971.1), *Oryza sativa* OsbHLH66 (XP_015627343.1), *Arabidopsis thaliana* AtbHLH66 (BAD44153.1) and *Malus domestica* MdbHLH66 (XP_028964276.1). The bHLH domain is marked with black solid bars. (**B**) Phylogenetic analysis of PtrbHLH66 and its homologous proteins from different plant species. (**C**) Analysis of the conserved motifs of PtrbHLH66 and its homologous proteins. Ten different motifs were identified and indicated by different colors. The nucleotide sequence of each conserved domain was also presented.

**Figure 2 ijms-23-15053-f002:**
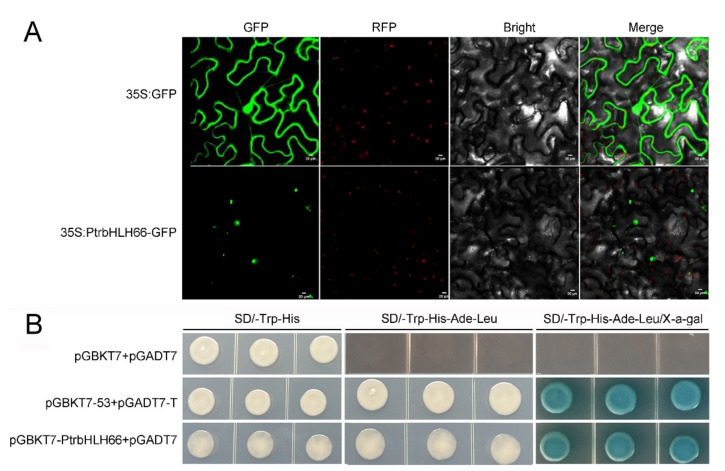
Subcellular localization and transcriptional activation analysis of PtrbHLH66. (**A**) Subcellular localization of PtrbHLH66 protein. The fluorescence signals of 35S:GFP (control) and 35S:PtrbHLH66-GFP proteins in epidermal cells of Nicotiana benthamiana leaves were observed with confocal microscopy. The confocal images were taken under GFP, RFP, bright light and merged fields, respectively. Bars represent 20 μm. (**B**) Transcriptional activation activity analysis of PtrbHLH66. The CDS sequence of PtrbHLH66 was introduced into the pGBKT7 vector to generate the fusion plasmid of pGBKT7-PtrbHLH66. Two pairs of plasmids (pGBKT7 and pGADT7, pGBKT7-53 and pGADT7-T) were co-transformed into Y2HGold yeast cells and used as negative control and positive control, respectively. Yeast cells were grown on SD/-Trp-His, SD/-Trp-His-Ade-Leu, and SD/-Trp-His-Ade-Leu/X-a-gal media.

**Figure 3 ijms-23-15053-f003:**
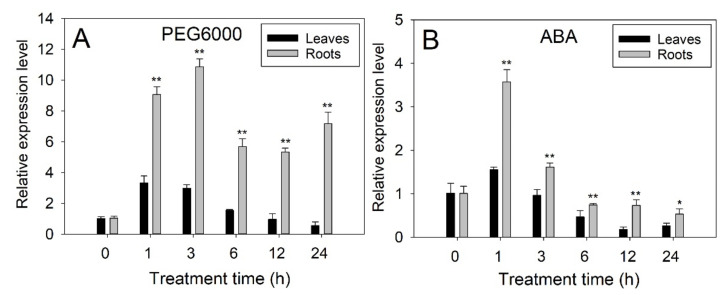
The expression analysis of *PtrbHLH66* in leaves and roots of trifoliate orange after (**A**) drought (20% PEG6000) and (**B**) ABA (100 μM) treatments. The y-axis records the relative gene expression levels as calculated by the 2^−ΔΔCT^ method with trifoliate orange *ACTIN* gene as the endogenous reference. Data are shown as the means ± SD calculated from three biological replicates. Significant expression differences between the leaves and roots at the *p* < 0.05 and *p* < 0.01 levels are indicated by single (*) and double asterisks (**), respectively, according to Student’s *t*-test.

**Figure 4 ijms-23-15053-f004:**
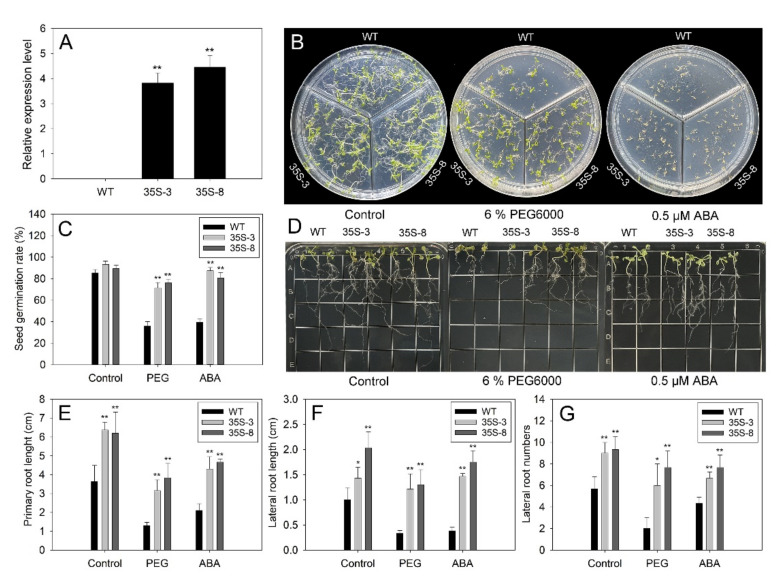
Drought and ABA responses of the transgenic Arabidopsis ectopic-expressing *PtrbHLH66* during germination and post-germination growth. (**A**) Expression of *PtrbHLH66* in the wild-type (WT) and two transgenic Arabidopsis lines (35S-3 and 35S-8). Analysis of (**B**) seed germination condition, (**C**) seed germination rates, (**D**) root growth condition, (**E**) primary root lengths, (**F**) lateral root lengths and (**G**) lateral root numbers of the two genotypes grown on MS medium containing 6% PEG6000 and 0.5 μM ABA. All data are shown as the means ± SD calculated from three biological replicates. Asterisks indicate that the values in transgenic Arabidopsis are significantly different from that of the WT plants (** *p* < 0.01, * *p* < 0.05).

**Figure 5 ijms-23-15053-f005:**
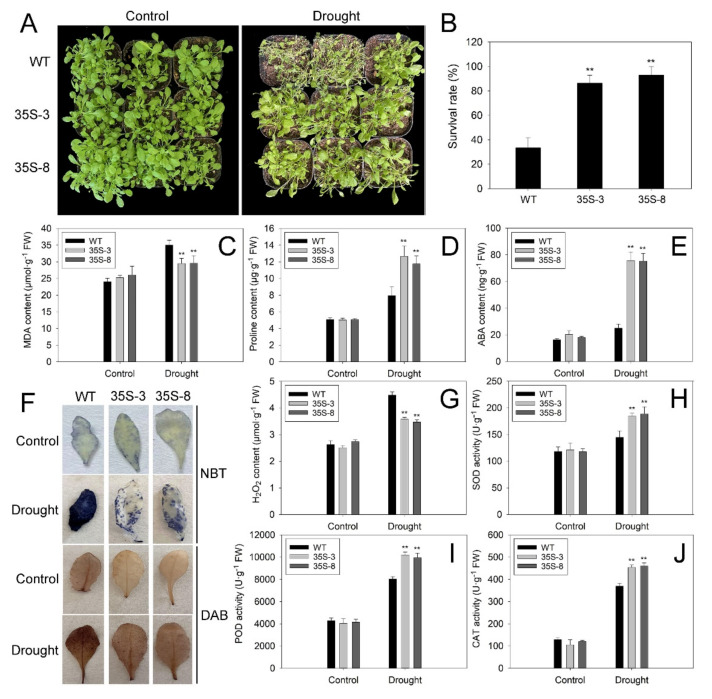
Drought tolerance analysis of the transgenic Arabidopsis ectopic-expressing *PtrbHLH66*. (**A**) Phenotype, (**B**) survival rates, (**C**) leaf MDA, (**D**) proline and (**E**) ABA contents, (**F**) leaf NBT and DAB staining, (**G**) H_2_O_2_ contents, (**H**) SOD, (**I**) POD and (**J**) CAT activities of the wild-type (WT) and transgenic seedlings were investigated under normal growth condition (control) and drought (withheld from watering for 3 weeks) stress. All data are shown as the means ± SD calculated from three biological replicates. Asterisks indicate that the values in transgenic Arabidopsis lines are significantly different from that of the WT plants (** *p* < 0.01).

**Figure 6 ijms-23-15053-f006:**
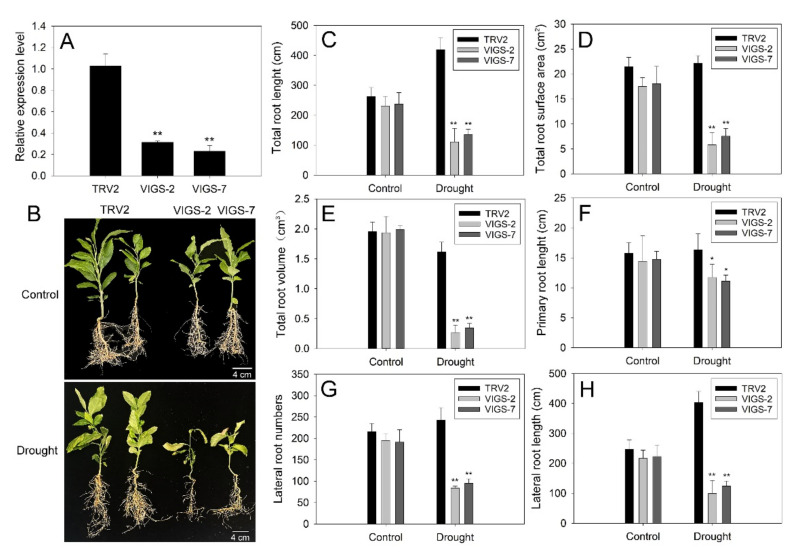
Drought tolerance analysis of *PtrbHLH66* homolog silencing lemon plants. (**A**) Expression of *PtrbHLH66* homolog in the TRV2 and two VIGS lemon lines (VIGS-2 and VIGS-7). Comparison of (**B**) phenotype and (**C**) total root lengths, (**D**) total root surface areas, (**E**) total root volumes, (**F**) primary root lengths, (**G**) lateral root numbers and (**H**) lateral root lengths between the TRV2 and two VIGS lemon lines under normal growth condition (control) and drought (withheld from watering for 3 weeks) stress. All data are shown as the means ± SD calculated from three biological replicates. Asterisks indicate that the values in the two VIGS lemon lines are significantly different from that of the TRV2 plants (** *p* < 0.01, * *p* < 0.05).

**Figure 7 ijms-23-15053-f007:**
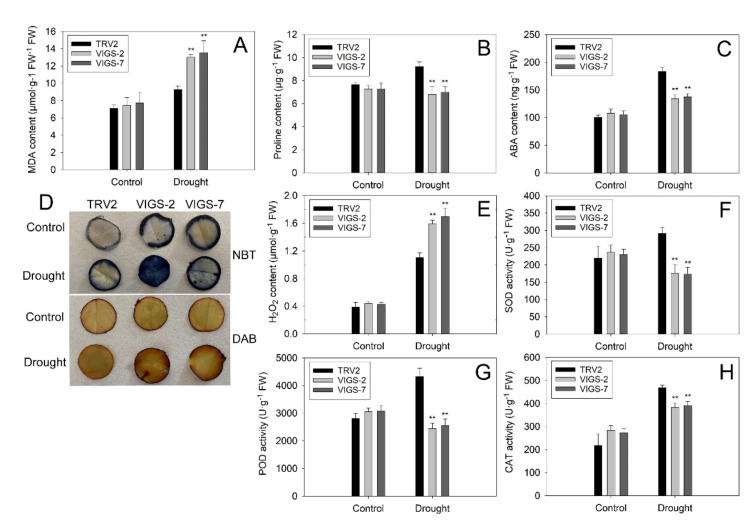
Comparison of physiological changes between the TRV2 and two *PtrbHLH66* homolog silencing lemon lines (VIGS-2 and VIGS-7) under normal growth condition (control) and drought (withheld from watering for 3 weeks) stress. The leaves of the two genotypes were collected to measure the (**A**) MDA, (**B**) proline and (**C**) ABA contents, (**D**) NBT and DAB staining, (**E**) H_2_O_2_ contents, (**F**) SOD, (**G**) POD and (**H**) CAT activities. All data are shown as the means ± SD calculated from three biological replicates. Asterisks indicate that the values in the two VIGS lemon lines are significantly different from that of the TRV2 plants (** *p* < 0.01).

**Figure 8 ijms-23-15053-f008:**
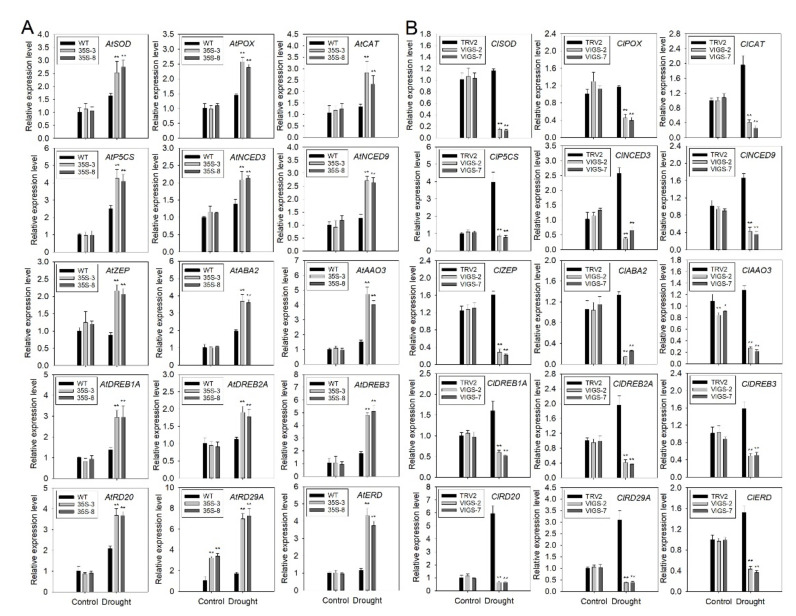
The expression analysis of drought-related genes in (**A**) two transgenic Arabidopsis lines (35S-3 and 35S-8) ectopic-expressing *PtrbHLH66* and (**B**) two *PtrbHLH66* homolog silencing lemon lines (VIGS-2 and VIGS-7) under normal growth condition (control) and drought (withheld from watering for 3 weeks) stress. The y-axis records the relative gene expression levels as calculated by the 2^−△△CT^ method. Data are shown as the means ± SD calculated from three biological replicates. Significant expression differences between the control and transgenic plants at the *p* < 0.05 and *p* < 0.01 levels are indicated by single (*) and double asterisks (**), respectively, according to Student’s *t*-test.

## Data Availability

The nucleotide sequence of *PtrbHLH66* has been submitted to the NCBI database with GenBank accession No. OP009586.

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
