# Peer review of "A Novel bHLH Transcription Factor PtrbHLH66 from Trifoliate Orange Positively Regulates Plant Drought Tolerance by Mediating Root Growth and ROS Scavenging"

_ijms, 2022, doi:10.3390/ijms232315053_

Round 1

Reviewer 1 Report

This study reported ectopic expression PtrbHLH66 positively regulated plant resistance to drought stress by regulating root growth and ROS scavenging. The results of this article is novel in scientific view.

However, there are a number of issues that need to be addressed.

1.In this article, why not use the gene knockout lines but the VIGS gene silencing lines?

2.Only two strain for OE and VIGS was used, which is not enough to support the conclusion. In addition, how the expression of ptrbHLH66 is affected in OE is not clear.

3.The transcript levels of genes involved in ABA biosynthesis, proline biosynthesis, ROS scavenging and drought response were obviously upregulated or downregulated in ectopic expressing lines. Whether these genes have binding sites for PtrbHLH66, or any interaction between them?

Author Response

Dear reviewer:

Thank you very much for your comment. We revised the manuscript in accordance with the reviewers’ comments, and carefully proof-read the manuscript to minimize typographical, grammatical, and bibliographical errors. Here below is our description on revision according to the reviewers’ comments.

Point-by-point answers to reviewer #1:

This study reported ectopic expression PtrbHLH66 positively regulated plant resistance to drought stress by regulating root growth and ROS scavenging. The results of this article is novel in scientific view. However, there are a number of issues that need to be addressed.

1.In this article, why not use the gene knockout lines but the VIGS gene silencing lines?

The authors’ Answer: Thank you for the reviewer’s comment. Virus-induced gene silencing (VIGS) is a natural defense response against the intrusion of exogenous nucleic acid in plants, which has been exploited as a reverse genetics tool for gene function analysis in diverse plant species.

   Compared with the stable transformation, the VIGS approach has a number of advantages. For example, the time involved in cloning the gene of interest and analysis of the VIGS phenotype can be done within 2-3 weeks, without going through the laborious and time-consuming process of generating stable mutants. VIGS can be applied to mature plants and thus allows the ability to assess the function for genes, whose mutation might be lethal in sexually propagated plants. Consequently, this facile technology is particularly important for plants with long life cycles, such as fruit trees.

Recently, the VIGS system is successfully used to identify the function of genes related to abiotic stress in citrus. For example, the VIGS system was employed to silence FcWRKY40

in kumquat and successfully proved that silencing of FcWRKY40 promotes salt susceptibility in kumquat (Dai et al., 2018; New phytol. 2018, 219(3):972-989, doi:10.1111/nph.15240); Song et al., (2022) silenced PtrABF2 in trifoliate orange by VIGS and proved that silencing of PtrABF2 inhibited putrescine synthesis and elevated drought susceptibility (Song et al., 2022; New phytol. doi:10.1111/nph.18526 ).

    Based on above reasons, we used VIGS gene silencing lines to identify the function of PtrbHLH66 in this paper.

2.Only two strain for OE and VIGS was used, which is not enough to support the conclusion. In addition, how the expression of ptrbHLH66 is affected in OE is not clear.

The authors’ Answer: Thank you for the reviewer’s comment. We agreed that three or more strains will support our conclusion better than two strain. However, to our knowledge, although three or more strain were used to characterize the plant gene function in some reports, there also many reports used two OE, one or two VIGS lines to identify the gene function in plants. For example, two transgenic tobacco and lemon lines overexpressing FcWRKY40 were used to confirm that FcWRKY40 can enhances salt tolerance in tobacco and lemon. One VIGS kumquat line was employed to prove that Silencing of FcWRKY40 causes hypersensitivity to salt stress (Dai et al., 2018; New phytol. 219(3): 972-989, doi:10.1111/nph.15240); Two lemon transgenic lines overexpressing PtrABF2 were used to prove that PtrABF2 improves drought tolerance by facilitating putrescine synthesis. One VIGS trifoliate orange line was employed to confirm the function of PtrABF2 in regulating putrescine synthesis and drought tolerance (Song et al., 2022; New phytol. doi:10.1111/nph.18526); Two transgenic trifoliate orange lines overexpressing PtrCDPK10 were used to prove that PtrCDPK10 enhances dehydration tolerance in transgenic trifoliate orange (Meng et al., 2019; Plant Sci. 2019, 291, 110320. doi:10.1016/j.plantsci.2019. 110320.) Liu et al., (2015) used two transgenic Arabidopsis lines to prove that AtbHLH112 confers ABA and abiotic stress tolerance to transgenic plants (Liu et al., 2015; New Phytol. 207(3):692–709. doi:10.1111/nph.13387). All above, we think two strains for OE and VIGS were enough to support the conclusion. However, the reviewer’s suggestion is very helpful. We will use three or more strains to characterize gene function in future studies.

     As shown in Fig. 4A, the expression of PtrbHLH66 in two positive OE lines (35S-3 and 35S-8) were significantly higher than the WT plants. We have added this statement in the revised manuscript (Line 180-182).

3.The transcript levels of genes involved in ABA biosynthesis, proline biosynthesis, ROS scavenging and drought response were obviously upregulated or downregulated in ectopic expressing lines. Whether these genes have binding sites for PtrbHLH66, or any interaction between them?

The authors’ Answer: Thank you for the reviewer’s comment. As the 15 genes were upregulated in the overexpressing lines but downregulated in the VIGS lines, we speculate that these genes may be directly or indirectly regulated by PtrbHLH66. In fact, we have analyzed the promoter of the 15 genes and found several E-box or G-box to which PtrbHLH66 might be bound. However, whether the PtrbHLH66 could bind to the E-box or G-box on the promoter of these genes needed further study using yeast one-hybrid, dual luciferase reporter assay system and electrophoresis mobility shift assay. In addition, PtrbHLH66 also might interacted with the proteins encoded by these genes, which should be investigated by bimolecular fluorescence complementation and co-immunoprecipitation. Next, we will further study the transcriptional regulation mechanism of PtrbHLH66 in a new paper.

    In the end of section 3 “Discussion”,we have added the discussion of this problem in the revised manuscript (Line 435-437).

The revisions in the revised manuscript are listed as follows:

  1. Line 37-41: According to the reviewer’s comment, we revised this sentence in the revised manuscript.
  2. Line 62, 64, 127-129, 156, 159, 181, 182, 207, 211-214, 219, 220, 236, 237: According to the reviewer’s comment, we have checked carefully throughout the manuscript to make sure first introduce the full name of abbreviation in the revised manuscript.
  3. Line 72-73: According to the reviewer’s comment, we have added the statistics of losses to citrus due to drought in the revised manuscript.
  4. Line 74, 111, 326, 327: We made the first letter of plant latin name capitalized.
  5. Line 119-121, 326-328: According to the reviewer’s comment, we have used the italicized form of sci names in the revised manuscript.
  6. Line 82-84: According to the reviewer’s comment, we have added many drought tolerance regulating genes/TFs, such as PtrABF, PtrMAPK, PtADC, PtsrMYB, PtrNAC72, PtrZPT-1 and PtrCDPK10, in the revised manuscript.
  7. Line 84-89: According to the reviewer’s comment, we have added the reason that we selected PtrbHLH66 for further study in the Section 1 of the revised manuscript.
  8. Line 108: According to the reviewer’s comment, we added the “with Poisson model” in this sentence of the revised manuscript.
  9. Line 180-182: According to the reviewer’s comment, we compared the expression levels of PtrbHLH66 in two positive transgenic lines and WT plant in the revised manuscript.
  10. Line 229: We used H2O2 to replace H2O2 in this sentence of the revised manuscript.
  11. Line 234: We added a word “homolog” and the latin name of lemon in this sentence of the revised manuscript.
  12. Line 240-241: According to the reviewer’s comment, we have deleted the sentence in Line 238-239 and added the sentence of “The control lemon plants were transformed with TRV2 empty vector and named TRV2.” in the revised manuscript.
  13. Line 357: We used the italicized form of “PtrHLH66” to replace “PtrHLH66”.
  14. Line 416, 417, 452, 453, 478, 525, 530, 531, 561-565: We deleted the full name of the abbreviation, because the full name appeared in the former part in the revised manuscript.
  15. Line 424: According to the reviewer’s comment, we added the sentence of “In addition, the expression of PtrbHLH66 was induced by ABA” in the revised manuscript.
  16. Line 425: We added a word “might”, and used a word “play” to replace “played” in the revised manuscript.
  17. Line 435-437: According to the reviewer’s comment, we have added the discussion of further research contents in the revised manuscript.
  18. Line 440-441: According to the reviewer’s comment, we have added the origin of the trifoliate orange seeds in the revised manuscript.
  19. Line 446-448: According to the reviewer’s comment, we have added the amount of irrigation water and the composition of the relevant growing substrate.
  20. Line 455-456: According to the reviewer’s comment, we have added the leaf age and position within the phylotax.
  21. Line 496-497, 506-509, 530: We revised the format of the name of restriction endonuclease.
  22. Line 701-703: We added one reference in the revised manuscript.
  23. Line 720-738: We added 7 references in the revised manuscript.

Reviewer 2 Report

Overall, the manuscript is well-written and presented in a scientific way.

On which bases do authors use 15 genes for qRT-PCR analysis? 

Author Response

Dear reviewer:

Thank you very much for your comment. We revised the manuscript in accordance with the reviewers’ comments, and carefully proof-read the manuscript to minimize typographical, grammatical, and bibliographical errors. Here below is our description on revision according to the reviewers’ comments.

Point-by-point answers to reviewer #2:

Overall, the manuscript is well-written and presented in a scientific way.

On which bases do authors use 15 genes for qRT-PCR analysis?

The authors’ Answer: Thank you for the reviewer’s comment. We selected some genes based on the corresponding physiological indexes detected in this paper. For example, our results showed that the activities of three antioxidant enzymes (SOD, POD and CAT) were remarkably increased in the OE lines, but obviously decreased in the VIGS lines after drought treatment. Thus, we selected three antioxidant enzyme genes (SOD, POX and CAT) for qRT-PCR analysis. After drought treatment, the two OE lines exhibited much higher levels of proline and ABA compared with the WT plants. While, the two VIGS lines showed the opposite trends. Thus, genes involved in ABA biosynthesis (AAO3, ABA2, NCED3, NCED9, ZEP) and proline biosynthesis (P5S5) were selected for qRT-PCR analysis.

The other genes, including DREB1A, DREB2A, DREB3, RD20, RD29, EDR1, are involved in plant drought response and often used for qRT-PCR analysis in transgenic plants overexpressing drought responsive bHLH TFs, such as buckwheat FtbHLH3 (Yao et al., 2017; Front. Plant Sci. 8:625. doi: 10.3389/fpls.2017.00625), apple MdbHLH130 (Zhao et al., 2020; Front. Plant Sci. 11:543696. doi: 10.3389/fpls.2020.543696), MdCIB1 (Ren et al., 2021; Environ. Exp. Bot. 188:104523. doi:10.1016/j.envexpbot.2021.104523).

The revisions in the revised manuscript are listed as follows:

  1. Line 37-41: According to the reviewer’s comment, we revised this sentence in the revised manuscript.
  2. Line 62, 64, 127-129, 156, 159, 181, 182, 207, 211-214, 219, 220, 236, 237: According to the reviewer’s comment, we have checked carefully throughout the manuscript to make sure first introduce the full name of abbreviation in the revised manuscript.
  3. Line 72-73: According to the reviewer’s comment, we have added the statistics of losses to citrus due to drought in the revised manuscript.
  4. Line 74, 111, 326, 327: We made the first letter of plant latin name capitalized.
  5. Line 119-121, 326-328: According to the reviewer’s comment, we have used the italicized form of sci names in the revised manuscript.
  6. Line 82-84: According to the reviewer’s comment, we have added many drought tolerance regulating genes/TFs, such as PtrABF, PtrMAPK, PtADC, PtsrMYB, PtrNAC72, PtrZPT-1 and PtrCDPK10, in the revised manuscript.
  7. Line 84-89: According to the reviewer’s comment, we have added the reason that we selected PtrbHLH66 for further study in the Section 1 of the revised manuscript.
  8. Line 108: According to the reviewer’s comment, we added the “with Poisson model” in this sentence of the revised manuscript.
  9. Line 180-182: According to the reviewer’s comment, we compared the expression levels of PtrbHLH66 in two positive transgenic lines and WT plant in the revised manuscript.
  10. Line 229: We used H2O2 to replace H2O2 in this sentence of the revised manuscript.
  11. Line 234: We added a word “homolog” and the latin name of lemon in this sentence of the revised manuscript.
  12. Line 240-241: According to the reviewer’s comment, we have deleted the sentence in Line 238-239 and added the sentence of “The control lemon plants were transformed with TRV2 empty vector and named TRV2.” in the revised manuscript.
  13. Line 357: We used the italicized form of “PtrHLH66” to replace “PtrHLH66”.
  14. Line 416, 417, 452, 453, 478, 525, 530, 531, 561-565: We deleted the full name of the abbreviation, because the full name appeared in the former part in the revised manuscript.
  15. Line 424: According to the reviewer’s comment, we added the sentence of “In addition, the expression of PtrbHLH66 was induced by ABA” in the revised manuscript.
  16. Line 425: We added a word “might”, and used a word “play” to replace “played” in the revised manuscript.
  17. Line 435-437: According to the reviewer’s comment, we have added the discussion of further research contents in the revised manuscript.
  18. Line 440-441: According to the reviewer’s comment, we have added the origin of the trifoliate orange seeds in the revised manuscript.
  19. Line 446-448: According to the reviewer’s comment, we have added the amount of irrigation water and the composition of the relevant growing substrate.
  20. Line 455-456: According to the reviewer’s comment, we have added the leaf age and position within the phylotax.
  21. Line 496-497, 506-509, 530: We revised the format of the name of restriction endonuclease.
  22. Line 701-703: We added one reference in the revised manuscript.
  23. Line 720-738: We added 7 references in the revised manuscript.

Reviewer 3 Report

The submitted manuscript deals with transcription factors that influence plant drought tolerance. As a model plant was used Poncirus trifoliata (L.) Raf.This is an interesting topic, but with regard to the cultivation of the mentioned type of plants on a limited area, the use of the results is somewhat debatable. I see the actual contribution of the manuscript in obtaining the relevant transcription factors and in the methodology. The manuscript is written relatively carefully using mainly newer literary sources. Results are clearly described and quantified. I only recommend enlarging or editing the graphs, as they are harder to read at this size. Perhaps it would be worth focusing on other analyzes or on mutual correlations. The discussion is rather descriptive in places. The methodology deserves addition. What was the origin of the seed? The height of the plants is given in the text, can the relevant BBCH phase be supplemented with data? I recommend specifying the amount of irrigation water and the composition of the relevant growing substrate. Explain the term young leaf. I recommend listing its position within the phylotax. In what form was ABA applied? PEG was used to induce drought stress. In this case, it is not drought stress, but osmotic stress. Why was PEG used and not e.g. irrigation limited?

Author Response

Dear reviewer:

Thank you very much for your comment. We revised the manuscript in accordance with the reviewers’ comments, and carefully proof-read the manuscript to minimize typographical, grammatical, and bibliographical errors. Here below is our description on revision according to the reviewers’ comments.

Point-by-point answers to reviewer #3:

The submitted manuscript deals with transcription factors that influence plant drought tolerance. As a model plant was used Poncirus trifoliata (L.) Raf. This is an interesting topic, but with regard to the cultivation of the mentioned type of plants on a limited area, the use of the results is somewhat debatable. I see the actual contribution of the manuscript in obtaining the relevant transcription factors and in the methodology. The manuscript is written relatively carefully using mainly newer literary sources.

The authors’ Answer: Thank you for the reviewer’s comment. The poncirus trifoliata is one of the most commonly used citrus rootstocks in China and also used in many other countries. The poncirus trifoliata was often used to study the function of drought responsive gene/TFs in citrus. For example, PtrABF, PtrMAPK, PtADC, PtsrMYB, PtrNAC72, PtrZPT-1, PtrCDPK10 and PtrABF2 have been reported to involved in poncirus trifoliata response to drought stress (Huang et al., 2010, BMC Plant Biol. 10: 230. doi:10.1186/1471-2229-10-230;Huang et al., 2011, J. Exp. Bot. 62, 5191–5206. doi:10.1093/jxb/ err229; Wang et al., 2011, J. Exp. Bot. 2011, 62, 2899–2914. doi:10.1093/jxb/erq463; Sun et al., 2014, Plant Physiol. Bioch. 78, 71–79. doi:10.1016/j.plaphy.2014.02.022; Wu et al., 2016, Plant Physiol. 172, 1532-1547. doi:10.1104/pp.16.01096; Liu et al., 2017, Plant Sci. 2017, 263, 66–78. doi:10.1016/j. plantsci.2017.07.012; Meng et al., 2019 Plant Sci. 2019, 291, 110320. doi:10.1016/j.plantsci. 2019.110320; Song et al., 2022, New phytol. doi:10.1111/nph.18526).

Results are clearly described and quantified. I only recommend enlarging or editing the graphs, as they are harder to read at this size.

The authors’ Answer: Thank you for the reviewer’s comment. We constructed the graphs based on the pixel request by the Journal and did our best to enlarge the font of the words in the graphs. The original graphs are large and clear enough to read. It is small in the manuscript because the graph quality severally declined after pasting in the manuscript.

Perhaps it would be worth focusing on other analyzes or on mutual correlations. The discussion is rather descriptive in places.

The authors’ Answer: Thank you for the reviewer’s comment. In the discussion section, we focused on discussing the correlations among physiological indices, gene expression and drought tolerance of transgenic plants (Line 360-437 in the revised manuscript). For example, we discussed the relationships among ROS accumulation, ROS scavenging enzyme activity (SOD, POD and CAT), corresponding gene expression and drought tolerance of transgenic plants. And we conclude that PtrbHLH66 positively regulated the transcript of antioxidant enzyme genes to enhance antioxidant enzyme activities, leading to improvement of ROS scavenging capacity and decrease of oxidative injury under drought stress (Line 371-390). Similarly, we discussed the relationship between root and drought tolerance of transgenic plants (Line 360-370), the correlation among proline content, the expression of proline biosynthesis gene and drought tolerance of transgenic plants (Line 391-402), the relationship among ABA level, the expression of ABA biosynthesis genes and drought tolerance of transgenic plants (Line 403-426), the relationship between drought responsive TFs/genes and drought tolerance of transgenic plants (Line 427-437).

According to the reviewer’s comment, we added the sentence of “In addition, the expression of PtrbHLH66 was induced by ABA” (Line 424) in the discussion of relationship among ABA level, the expression of ABA biosynthesis genes and drought tolerance of transgenic plants to further suggest that PtrbHLH66 might play an important role in regulating plant drought tolerance by ABA-dependent pathway.

The methodology deserves addition. What was the origin of the seed? The height of the plants is given in the text, can the relevant BBCH phase be supplemented with data? I recommend specifying the amount of irrigation water and the composition of the relevant growing substrate.

The authors’ Answer: Thank you for the reviewer’s comment. The seeds were obtained from Xinfeng County, Ganzhou City, Jiangxi province in China. We are sorry that we cannot found the BBCH standard for Poncirus trifoliata, but the seedlings were cultivated for three months. Germinated seeds were transplanted to pots filled with soil (garden soils, peats and sands mixed at the ratio of 3: 2: 1), irrigated with 200 ml water every three days, fertilized by compound fertilizer (N ≥ 15 %, K2O ≥15 %, P2O5 ≥ 15 %) every 15 days and grown to the height of 28±2 cm after three months cultivation.

    According to the reviewer’s comment, we added these statements in the revised manuscript (Line 440-448).

Explain the term young leaf. I recommend listing its position within the phylotax.

The authors’ Answer: Thank you for the reviewer’s comment. For extracted total RNA more efficiently, we selected the third and fourth young leaf (5 to 6 days old, not mature) within phylotax.

According to the reviewer’s comment, we added this statements in the revised manuscript (Line 455-456).

In what form was ABA applied?

The authors’ Answer: Thank you for the reviewer’s comment. We used ABA solution diluted to 100 µM in Hoagland’s solution.

PEG was used to induce drought stress. In this case, it is not drought stress, but osmotic stress. Why was PEG used and not e.g. irrigation limited?

The authors’ Answer: Thank you for the reviewer’s comment. PEG was often used to simulate drought stress (Shi et al., 2015, Environ exp bot, 111, 127–134. doi:10.1016/j.envexpbot.2014.11.008; Guo et al., 2020, Plant Physiol. and Bioch. 154: 85-93 doi:10.1016/j.plaphy.2020.06.008; Zhao et al., 2021, Int. J. Mol. Sci. 22, 3294.. doi:10.3390/ijms22073294). The expression levels of most genes encoding TF were induced or suppressed by drought within several hours, and then declined to a low level. The irrigation limited cannot reach the drought effect within several hours. Thus, we investigated the expression of PtrbHLH66 under PEG6000 to simulate drought stress.

The revisions in the revised manuscript are listed as follows:

  1. Line 37-41: According to the reviewer’s comment, we revised this sentence in the revised manuscript.
  2. Line 62, 64, 127-129, 156, 159, 181, 182, 207, 211-214, 219, 220, 236, 237: According to the reviewer’s comment, we have checked carefully throughout the manuscript to make sure first introduce the full name of abbreviation in the revised manuscript.
  3. Line 72-73: According to the reviewer’s comment, we have added the statistics of losses to citrus due to drought in the revised manuscript.
  4. Line 74, 111, 326, 327: We made the first letter of plant latin name capitalized.
  5. Line 119-121, 326-328: According to the reviewer’s comment, we have used the italicized form of sci names in the revised manuscript.
  6. Line 82-84: According to the reviewer’s comment, we have added many drought tolerance regulating genes/TFs, such as PtrABF, PtrMAPK, PtADC, PtsrMYB, PtrNAC72, PtrZPT-1 and PtrCDPK10, in the revised manuscript.
  7. Line 84-89: According to the reviewer’s comment, we have added the reason that we selected PtrbHLH66 for further study in the Section 1 of the revised manuscript.
  8. Line 108: According to the reviewer’s comment, we added the “with Poisson model” in this sentence of the revised manuscript.
  9. Line 180-182: According to the reviewer’s comment, we compared the expression levels of PtrbHLH66 in two positive transgenic lines and WT plant in the revised manuscript.
  10. Line 229: We used H2O2 to replace H2O2 in this sentence of the revised manuscript.
  11. Line 234: We added a word “homolog” and the latin name of lemon in this sentence of the revised manuscript.
  12. Line 240-241: According to the reviewer’s comment, we have deleted the sentence in Line 238-239 and added the sentence of “The control lemon plants were transformed with TRV2 empty vector and named TRV2.” in the revised manuscript.
  13. Line 357: We used the italicized form of “PtrHLH66” to replace “PtrHLH66”.
  14. Line 416, 417, 452, 453, 478, 525, 530, 531, 561-565: We deleted the full name of the abbreviation, because the full name appeared in the former part in the revised manuscript.
  15. Line 424: According to the reviewer’s comment, we added the sentence of “In addition, the expression of PtrbHLH66 was induced by ABA” in the revised manuscript.
  16. Line 425: We added a word “might”, and used a word “play” to replace “played” in the revised manuscript.
  17. Line 435-437: According to the reviewer’s comment, we have added the discussion of further research contents in the revised manuscript.
  18. Line 440-441: According to the reviewer’s comment, we have added the origin of the trifoliate orange seeds in the revised manuscript.
  19. Line 446-448: According to the reviewer’s comment, we have added the amount of irrigation water and the composition of the relevant growing substrate.
  20. Line 455-456: According to the reviewer’s comment, we have added the leaf age and position within the phylotax.
  21. Line 496-497, 506-509, 530: We revised the format of the name of restriction endonuclease.
  22. Line 701-703: We added one reference in the revised manuscript.
  23. Line 720-738: We added 7 references in the revised manuscript.

Reviewer 4 Report

The manuscript ijms-2002920 reports the functional characterization of a bHLH TF from trifoliate orange. The authors report that this TF is involved in drought stress tolerance. It functions so by scavenging ROS and mediating root growth.

L37. The sentence is long and grammatically incorrect.

You must first introduce the abbreviation and then use it. E.g., you used bHLH first but introduced its full name in the second paragraph. This should be checked carefully throughout the ms.

L69 and onward. Authors should include statistics of losses to citrus due to drought.

L74. Authors should introduce the sci name of trifoliate orange here instead of in L76.

Are there any other known drought tolerance regulating genes/TFs in this species?

Section 2.1. Authors should include data/details which lead them to select this specific TF for this project. Was there any preliminary data from which you identified this as a highly expressed TF in response to drought? what is the basis of your choice of this TF?

L97-98. Which model was used for construct the tree?

Section 2.1. What do these results indicate? Your presentation here was good for the data but what does this data mean?

At some place PtrbHLH66 is italicized and in other text it isn’t.

L224-25 is a confusing statement.

L311 and onwards. The sci names are not italicized.

L327-8. Does this study confirm that the characterized TF produces drought responses in trifoliate citrus in an ABA-dependent way? I think it is premature data to state that. We do not know now that if the same works in ABA-independent way. The author wrote it as a suggestion; thus, they must include a further statement that it needs to be further explored if it works only in ABA-dependent ways or both ways.

Author Response

Dear reviewer:

Thank you very much for your comment. We revised the manuscript in accordance with the reviewers’ comments, and carefully proof-read the manuscript to minimize typographical, grammatical, and bibliographical errors. Here below is our description on revision according to the reviewers’ comments.

The manuscript ijms-2002920 reports the functional characterization of a bHLH TF from trifoliate orange. The authors report that this TF is involved in drought stress tolerance. It functions so by scavenging ROS and mediating root growth.

L37. The sentence is long and grammatically incorrect.

The authors’ Answer: Thank you for the reviewer’s comment, we have revised this sentence in the revised manuscript (Line 37-41).

You must first introduce the abbreviation and then use it. E.g., you used bHLH first but introduced its full name in the second paragraph. This should be checked carefully throughout the ms.

The authors’ Answer: Thank you for the reviewer’s comment, we have checked carefully throughout the manuscript to make sure first introduce the full name of abbreviation in the revised manuscript (Line 62, 64, 127-129, 156, 159, 181, 182, 207, 211-214, 219, 220, 236, 237).

L69 and onward. Authors should include statistics of losses to citrus due to drought.

The authors’ Answer: Thank you for the reviewer’s comment, we have added the statistics of losses to citrus due to drought in the revised manuscript (Line 72-73).

L74. Authors should introduce the sci name of trifoliate orange here instead of in L76.

The authors’ Answer: Thank you for the reviewer’s comment, we have introduced the sci name of trifoliate orange in Line 77 instead of in Line 79 in the revised manuscript.

Are there any other known drought tolerance regulating genes/TFs in this species?

The authors’ Answer: Thank you for the reviewer’s comment. In fact, there are many drought tolerance regulating genes/TFs, such as PtrABF, PtrMAPK, PtADC, PtsrMYB, PtrNAC72, PtrZPT-1 and PtrCDPK10, have been identified in trifoliate orange. We have added these genes/TFs in the revised manuscript (Line 82-84).

Section 2.1. Authors should include data/details which lead them to select this specific TF for this project. Was there any preliminary data from which you identified this as a highly expressed TF in response to drought? what is the basis of your choice of this TF?

The authors’ Answer: Thank you for the reviewer’s comment. Recently, we performed transcriptome sequencing to screen differentially expressed genes (DEGs) from trifoliate orange under normal growth condition and drought stress. In these DEGs, the expression of a bHLH family gene, named PtrbHLH66, was significantly upregulated under drought stress, suggesting it might play an important role in trifoliate orange tolerance to drought stress. Thus, we further cloned and characterized PtrbHLH66 in this paper.

    The transcriptome data was prepared to published in another paper, thus we cannot show it in the Section 2.1. of this paper. However, we have added the reason that we selected PtrbHLH66 for further study in the Section 1 of the revised manuscript (Line 84-89).

L97-98. Which model was used for construct the tree?

The authors’ Answer: Thank you for the reviewer’s comment. We used the Poisson model of the neighbor-joining algorithm in MEGA X software to construct the phylogenetic tree. We have added this statement in the revised manuscript (Line 108).

Section 2.1. What do these results indicate? Your presentation here was good for the data but what does this data mean?

The authors’ Answer: Thank you for the reviewer’s comment. Section 2.1 first describe the basic information of PtrbHLH66 and its coding protein. The BLAST, multiple sequence alignment and MEME analyses showed that PtrbHLH66 shared high identities and conserved domains with other plant bHLH66 proteins, suggesting PtrbHLH66 belonged to bHLH family. The phylogenetic tree analysis revealed the phylogenetic relationship between PtrbHLH66 and other plant bHLH66 proteins.

L224-25 is a confusing statement.

The authors’ Answer: Thank you for the reviewer’s comment. This statement means the control plants of the VIGS lines was the wild type lemon transformed with TRV2 empty vector. We have changed this statement to “The control lemon plants were transformed with TRV2 empty vector and named TRV2.” in the revised manuscript (Line 240-241).

L311 and onwards. The sci names are not italicized.

The authors’ Answer: Thank you for the reviewer’s comment. We have used the italicized form of sci names in the revised manuscript (Line 119-121, 326-328).

L327-8. Does this study confirm that the characterized TF produces drought responses in trifoliate citrus in an ABA-dependent way? I think it is premature data to state that. We do not know now that if the same works in ABA-independent way. The author wrote it as a suggestion; thus, they must include a further statement that it needs to be further explored if it works only in ABA-dependent ways or both ways.

The authors’ Answer: Thank you for the reviewer’s comment. We agreed with the reviewer’s comment. We have added the statement in the revised manuscript (Line 435-437). 

The revisions in the revised manuscript are listed as follows:

  1. Line 37-41: According to the reviewer’s comment, we revised this sentence in the revised manuscript.
  2. Line 62, 64, 127-129, 156, 159, 181, 182, 207, 211-214, 219, 220, 236, 237: According to the reviewer’s comment, we have checked carefully throughout the manuscript to make sure first introduce the full name of abbreviation in the revised manuscript.
  3. Line 72-73: According to the reviewer’s comment, we have added the statistics of losses to citrus due to drought in the revised manuscript.
  4. Line 74, 111, 326, 327: We made the first letter of plant latin name capitalized.
  5. Line 119-121, 326-328: According to the reviewer’s comment, we have used the italicized form of sci names in the revised manuscript.
  6. Line 82-84: According to the reviewer’s comment, we have added many drought tolerance regulating genes/TFs, such as PtrABF, PtrMAPK, PtADC, PtsrMYB, PtrNAC72, PtrZPT-1 and PtrCDPK10, in the revised manuscript.
  7. Line 84-89: According to the reviewer’s comment, we have added the reason that we selected PtrbHLH66 for further study in the Section 1 of the revised manuscript.
  8. Line 108: According to the reviewer’s comment, we added the “with Poisson model” in this sentence of the revised manuscript.
  9. Line 180-182: According to the reviewer’s comment, we compared the expression levels of PtrbHLH66 in two positive transgenic lines and WT plant in the revised manuscript.
  10. Line 229: We used H2O2 to replace H2O2 in this sentence of the revised manuscript.
  11. Line 234: We added a word “homolog” and the latin name of lemon in this sentence of the revised manuscript.
  12. Line 240-241: According to the reviewer’s comment, we have deleted the sentence in Line 238-239 and added the sentence of “The control lemon plants were transformed with TRV2 empty vector and named TRV2.” in the revised manuscript.
  13. Line 357: We used the italicized form of “PtrHLH66” to replace “PtrHLH66”.
  14. Line 416, 417, 452, 453, 478, 525, 530, 531, 561-565: We deleted the full name of the abbreviation, because the full name appeared in the former part in the revised manuscript.
  15. Line 424: According to the reviewer’s comment, we added the sentence of “In addition, the expression of PtrbHLH66 was induced by ABA” in the revised manuscript.
  16. Line 425: We added a word “might”, and used a word “play” to replace “played” in the revised manuscript.
  17. Line 435-437: According to the reviewer’s comment, we have added the discussion of further research contents in the revised manuscript.
  18. Line 440-441: According to the reviewer’s comment, we have added the origin of the trifoliate orange seeds in the revised manuscript.
  19. Line 446-448: According to the reviewer’s comment, we have added the amount of irrigation water and the composition of the relevant growing substrate.
  20. Line 455-456: According to the reviewer’s comment, we have added the leaf age and position within the phylotax.
  21. Line 496-497, 506-509, 530: We revised the format of the name of restriction endonuclease.
  22. Line 701-703: We added one reference in the revised manuscript.
  23. Line 720-738: We added 7 references in the revised manuscript.
